# Burden, determinants, consequences and care of multimorbidity in rural and urbanising Telangana, India: protocol for a mixed-methods study within the APCAPS cohort

Judith Lieber ,[1] Santosh Kumar Banjara,[2] Poppy Alice Carson Mallinson ,[1] Hemant Mahajan,[2] Santhi Bhogadi,[3] Srivalli Addanki,[3] Nick Birk,[1] Wenbo Song ,[1,4] Anoop SV Shah,[5] Om Kurmi,[6] Gowri Iyer,[7] Sureshkumar Kamalakannan ,[8,9] Raghu Kishore Galla,[10] Shilpa Sadanand,[7] Teena Dasi,[2] Bharati Kulkarni ,[2,11] Sanjay Kinra[1]

JL and SKB contributed equally.

For numbered affiliations see end of article.

**Correspondence to**
Dr Judith Lieber;
judith.lieber1@lshtm.ac.uk

## ABSTRACT

**Introduction** The epidemiological and demographic transitions are leading to a rising burden of multimorbidity (co-occurrence of two or more chronic conditions) worldwide. Evidence on the burden, determinants, consequences and care of multimorbidity in rural and urbanising India is limited, partly due to a lack of longitudinal and objectively measured data on chronic health conditions. We will conduct a mixed-methods study nested in the prospective Andhra Pradesh Children and Parents' Study (APCAPS) cohort to develop a data resource for understanding the epidemiology of multimorbidity in rural and urbanising India and developing interventions to improve the prevention and care of multimorbidity.

**Methods and analysis** We aim to recruit 2100 APCAPS cohort members aged 45+ who have clinical and lifestyle data collected during a previous cohort follow-up (2010–2012). We will screen for locally prevalent non-communicable, infectious and mental health conditions, alongside cognitive impairments, disabilities and frailty, using a combination of self-reported clinical diagnosis, symptom-based questionnaires, physical examinations and biochemical assays. We will conduct in-depth interviews with people with varying multimorbidity clusters, their informal carers and local healthcare providers. Deidentified data will be made available to external researchers.

**Ethics and dissemination** The study has received approval from the ethics committees of the National Institute of Nutrition and Indian Institute of Public Health Hyderabad, India and the London School of Hygiene and Tropical Medicine, UK. Meta-data and data collection instruments will be published on the APCAPS website alongside details of existing APCAPS data and the data access process (www.lshtm.ac.uk/research/centres-projects-groups/apcaps).

## STRENGTHS AND LIMITATIONS OF THIS STUDY

⇒ Open-access multimorbidity resource of 2100 adults aged 45+ participating in the community-based Andhra Pradesh Children and Parents' Study cohort in rural and urbanising Telangana, India, with scope to conduct longitudinal analyses to examine biological, behavioural, sociodemographic and environmental risk factors for multimorbidity, multimorbidity progression and consequences.

⇒ A broad range of locally prevalent chronic conditions screened using objective measures to estimate the burden of multimorbidity, overcoming the limitations of self-reporting.

⇒ A qualitative approach to explore the lived experience of people with multimorbidity and their informal carers and identify current gaps in care to guide intervention and policy development.

⇒ Statistical power may be insufficient to examine some less prevalent clusters of multimorbidity.

⇒ The study population is based in a defined geographical area that does not represent the entire population of India and may not capture multimorbidity clusters specific to other Indian settings.

## INTRODUCTION

Multimorbidity, defined as the co-occurrence of two or more conditions of long-duration ('chronic') in an individual, is rising globally.[1] Multimorbidity is associated with poor quality of life, disability, mortality and healthcare use, which significantly strain health systems.[1] In rural India, where 65% of the Indian population reside, socioeconomic developments and globalisation are introducing new risk factors for chronic conditions (eg, air pollution) alongside prevailing poverty-related risks (eg, undernutrition). The rising prevalence of non-communicable diseases alongside an ongoing burden of infectious disease is leading to a significant

rise in multimorbidity.[2 3] Although a system for primary care exists, the majority of outpatient health services are provided by the private sector.[4] Lack of a robust system of primary care leaves people with multimorbidity highly vulnerable to catastrophic health expenditure and poor health and social outcomes.[5 6] Evidence on the burden, determinants, consequences and gaps in care of multimorbidity is required to improve the prevention and care of multimorbidity in rural and urbanising India and other settings undergoing similar trends.[2]

Current evidence on the burden of multimorbidity in rural and urbanising India is limited. Incidence data are lacking, while prevalence estimates have fluctuated widely (ranging from 9% to 59%) owing to a frequent reliance on self-reported disease, which is known to grossly underestimate true disease prevalence in these settings.[6–9] There is some evidence that multimorbidity clusters in India mirror those reported in high-income countries, specifically, clustering of cardiorespiratory (angina, asthma, chronic obstructive pulmonary disease (COPD)), metabolic (diabetes, obesity, hypertension) and mental articular (arthritis and depression) conditions.[10–14] However, multimorbidity clusters specific to rural and urbanising India may have not been identified as several locally prevalent conditions (eg, anaemia) were not measured in prior studies.[9–11 13 14]

Longitudinal studies from high-income countries have identified several risk factors (eg, air pollution) associated with multiple chronic conditions (eg, cardiorespiratory, cancers) and social determinants of multimorbidity.[15 16] However, longitudinal evidence on the risk factors for multimorbidity is critically lacking in India and other low-income and middle-income countries.[17 18] While a few longitudinal studies from India have investigated the risk factors of individual chronic conditions, the effect of these or other common risk factors in rural India (eg, indoor use of biomass fuel) on multimorbidity has not been examined.[7]

Studies in high-income countries have reported the challenges of caring for multimorbidity for the individual as well as their informal carers (eg, impacted social relationships) and healthcare providers (eg, lack of clinical guidance for treatment of multiple conditions).[19–21] Studies of individual conditions in India have highlighted long, complex and burdensome care pathways.[22 23] However, evidence of the lived experience of multimorbidity for patients and their informal carers is lacking in India, particularly in rural areas. Further, few studies have explored the barriers to care of multimorbidity in India, focusing specifically on diabetes comorbidities.[22 24]

## AIM AND OBJECTIVES

We aim to develop a longitudinal data resource for examining the epidemiology of multimorbidity in rural and urbanising India and developing interventions to improve the prevention and care of multimorbidity. We envisage that this will also have relevance for other rural and urbanising settings at similar stages in their epidemiological transition. Our objectives are as follows:

1. To estimate the prevalence and incidence of multimorbidity (defined as two or more chronic conditions) and common multimorbidity clusters in rural and urbanising Telangana, India.
2. To examine the biological, behavioural, sociodemographic and environmental risk factors for, and consequences of, multimorbidity.
3. To explore the lived experience of individuals with multimorbidity and their informal carers.
4. To assess how care for multimorbidity is organised in rural and urbanising India and identify the barriers and enablers to high-quality care.

## METHODS AND ANALYSIS
### Study design
A mixed-methods study in the community-based Andhra Pradesh Children and Parents Study (APCAPS) cohort.[25] The qualitative methods (objectives 3–4) will be embedded within the quantitative study (objectives 1–2), using the quantitative data as a sampling frame.

### Study population and setting
APCAPS is a prospective intergenerational cohort that has been incrementally built through long-term follow-up of a nutrition trial (1987–1990) conducted in 29 villages (Ranga Reddy district) in the south Indian state of Telangana. The 29 villages of the APCAPS cohort are located 50–100 km from Hyderabad, the state capital and have urbanised at varying rates, now presenting a mix of rural and urban risk factors. For children born during the original nutrition trial, chronic disease risk factor and outcome data has been collected repeatedly (2003–2005 to 2010–2012). In 2010–2012, similar data were collected for their parents and siblings (N=6972 total cohort). Figure 1 provides an overview of the longitudinal multimorbidity analyses feasible with the 2010–2012 and 2022–2023 data.

### Inclusion criteria
Adults aged 45 years and above who participated in a prior APCAPS study residing in the study villages or surrounding area are eligible to participate. This group primarily comprises the parents' generation of the cohort plus older siblings of the trial children. While the target population are aged 45+, cohort members aged 18–44 are eligible to undergo a limited protocol if they wish to participate.

### Exclusion criteria
None.

### Quantitative data collection

### Quantitative sample
A cohort tracking exercise (April–August 2021) identified N=2571 APCAPS members eligible to participate

**Figure 1** Overview of feasible analyses examining the determinants, burden and consequences of multimorbidity using previously collected (2010–2012) and planned (2022–2023) APCAPS data. APCAPS, Andhra Pradesh Children and Parents' Study.

(58% women). We anticipate a final sample size of N=2057 (expected mean age 63 and 56 years for men and women, respectively) based on an estimated 80% follow-up. The proposed sample size will allow us to estimate the 10-year incidence of any multimorbidity (defined as two or more chronic conditions), and incidence of specific common multimorbidity clusters, with a precision of 4% and 3%, respectively (assuming a conservative N=700 new cases of multimorbidity and three equal-sized clusters).[10] For analysis of risk factors, we will have statistical power to detect a rate ratio of ~1.25 (assuming an incidence rate of 35/1000 person-years in the unexposed with a 30% risk factor prevalence and 80% power at a 5% significance level).

### Quantitative data collection procedures

Following piloting, data collection commenced in February 2022 and will be completed towards the end of 2023. Data collection is occurring village-by-village, with clinics occurring in the study villages or at the central study office (Indian Council of Medical Research (ICMR)-National Institute of Nutrition (NIN), Hyderabad), or the household when necessary, for example, for people with mobility impairments. Questionnaires and physical examinations occur at the clinic, and participants are invited to return the following weekend to provide blood and urine samples. Questionnaire responses and results of most physical assessments are collected by tablet using the Open Data Kit (ODK) platform.[26] Other data are extracted directly from the relevant instruments.

### Quantitative protocol

The full protocol consists of questionnaires, a physical examination and collection of urine and fasting blood samples. The screened conditions were selected based on their reported prevalence in India and their importance to the local community.[27–30] We are screening for chronic diseases with a combination of self-reported diagnosis by a healthcare professional, symptom-based questionnaires, physical examinations and biochemical assays, in order to overcome the socioeconomic bias observed when classifying conditions by prior clinical diagnosis in rural India[8] (table 1). The questionnaires cover sociodemographics, behavioural risk factors, self-reported clinical diagnosis of chronic conditions, treatment details, symptom-based screening tools and impact measures (eg, quality of life). We limit questions about prior clinical diagnosis to conditions with a higher likelihood of diagnosis in the study population, based on assessment of existing and pilot APCAPS data and discussions with local healthcare providers. The protocol was designed to maximise the scope for longitudinal analyses and to measure key chronic conditions and impact outcomes.[31]

The physical assessment includes a detailed examination of cardiovascular physiology, anthropometry and physical functioning assessments, and assessment of diabetes complications (neuropathy and ulceration) for participants reporting a prior diagnosis of diabetes or screening positive for pre-diabetes/diabetes. Individuals with limb or spine deformities are excluded from the anthropometry protocol, and individuals with mobility impairments are excluded from the 6 meter timed walk if they/the fieldworker deem it unsafe. Pregnant women and individuals with a prior diagnosis of diabetes are not requested to fast overnight for blood sampling but are eligible to provide a sample. As this is an ongoing cohort,

**Table 1** Screened health conditions and states of poor health, by type of screening measure

| Health condition/state | Screening measure | | |
| --- | --- | --- | --- |
| | Self-reported clinical diagnosis | Symptom-based questionnaire | Physical examination/ biochemical assay |
| Hypertension | ✔ | | ✔ |
| Left ventricular dysfunction/heart failure* | ✔ | | ✔ |
| Coronary heart disease* | ✔ | ✔ | |
| Atrial fibrillation* | ✔ | | ✔ |
| Atherosclerosis | | | ✔ |
| Stroke | ✔ | ✔ | |
| Underweight/obesity | | | ✔ |
| Diabetes | ✔ | | ✔ |
| Diabetic ulceration | ✔ | | ✔ |
| Diabetic neuropathy | ✔ | ✔ | ✔ |
| Peripheral arterial disease | ✔ | ✔ | ✔ |
| Retinopathy | ✔ | | ✔ |
| Cataracts | ✔ | ✔ | |
| Chronic obstructive pulmonary disease | | ✔ | |
| Asthma | ✔ | ✔ | |
| Sarcopenia | | | ✔ |
| Arthritis | ✔ | ✔ | |
| Anaemia | | | ✔ |
| Depression | | ✔ | |
| Anxiety | | ✔ | |
| Alcohol use disorder | | ✔ | |
| Dementia | | ✔ | |
| Tuberculosis | ✔ | ✔ | |
| COVID-19 | ✔ | ✔ | |
| Thyroid dysfunction | ✔ | ✔ | ✔ |
| Chronic liver disease | ✔ | | ✔ |
| Chronic kidney disease | ✔ | | ✔ |
| Peptic ulcer | ✔ | | |
| Gastro-oesophageal reflux disease | ✔ | ✔ | |
| Cancer | ✔ | | |
| Poor oral health | | ✔ | |
| Chronic pain | | ✔ | |
| Chronic stress | | ✔ | |
| Frailty | | ✔ | ✔ |
| Disability | | ✔ | ✔ |
| Vision impairment | ✔ | | ✔ |
| Hearing impairment | ✔ | | ✔ |

*Question on prior clinical diagnosis refers to 'heart disease' but does not categorise by condition.

we plan to prospectively collect mortality data, including cause of death.

## Validity of quantitative protocol

Hypertension is assessed by seated brachial blood pressure (three readings) with validated oscillometric devices[32–34] (table 2). Left ventricular dysfunction is assessed by two-dimensional (2D) echocardiography performed by an experienced echo technician following a standardised protocol,[35] coronary heart disease symptoms with the validated WHO Rose questionnaire,[36] atrial fibrillation with 12-lead ECG and atherosclerosis with Carotid–Intima Media Thickness (CIMT). A stroke is measured with a symptom-based questionnaire developed and commonly used to estimate the community-based prevalence in India.[37] Underweight and obesity are assessed with height, weight, and hip and waist circumferences to estimate body mass index (BMI) and waist-to-hip ratio, and bioelectrical impedance to estimate body fat percentage, with all measures taken twice and using the same standardised protocols as earlier APCAPS surveys.[25]

Diabetes is screened by glycated haemoglobin and fasting plasma glucose, both gold-standard diagnostic tests.[38] Participants with self-reported diabetes and/or who screen positive for pre-diabetes or diabetes (based on fasting plasma glucose) are undergoing a diabetic neuropathy examination and assessment for diabetic ulceration. We do not predict advanced neuropathy in the study population due to their relatively young age (average 59 years). As such, we are using the validated modified Toronto Clinical Neuropathy Score (mTCNS), which aims to detect early dysfunction.[39] Diabetic foot ulcerations are assessed with infrared thermal imaging, a validated measure.[40] In all participants, peripheral arterial disease (PAD) is assessed by toe brachial index, which has better sensitivity in people with diabetes (in whom we predict most cases of PAD) versus the more commonly used ankle brachial index.[41] Retinal images are being taken to screen for retinopathy, with two fields (centred on the optic disc and the macula) to improve validity.[42] We are using a non-mydriatic camera without pupil dilation to maximise participant comfort. Cataracts are assessed with a symptom-based tool previously used in a nationwide survey in India.[43]

To reduce risk of COVID-19 transmission, chronic respiratory conditions (COPD and asthma) are assessed with validated symptom-based questionnaires[44 45] instead of spirometry (used in prior APCAPS surveys[46]). Sarcopenia is defined by muscle strength, physical performance and appendicular skeletal muscle mass.[47] Muscle strength is measured by grip strength using the same instrument and protocol as the 2010–2012 APCAPS survey as grip strength estimates are highly sensitive to instrument and protocol used.[25 48] In line with the Asian Working Group for Sarcopenia's recommendations, physical functioning is being measured with a 6 meter timed walk at normal speed with a dynamic start, and appendicular skeletal muscle mass with bioimpedance analysis.[47] Arthritis is screened with

a cross-culturally valid symptom-based questionnaire.[49] Anaemia is assessed by haemoglobin concentration of fasting venous blood with an auto-analyser on the day of sample collection (a gold-standard diagnostic test[50]). Mental health conditions (depression, anxiety and alcohol-use disorder) and dementia are being screened with widely used questionnaires validated for Indian populations.[51–53] Due to the relatively low expected prevalence of tuberculosis in the study population, and high resource requirements and participant burden for diagnostic testing, we screen for tuberculosis with a bespoke symptom-based questionnaire. The questions are based on the Government of India's definition of presumptive tuberculosis and WHO guidance for tuberculosis surveys to allow within-country and across-country prevalence comparisons, respectively.[54–56] COVID-19 history and post-COVID-19 condition (Long COVID) are assessed with a combination of self-report and symptom-based questions devised for the study based on WHO case definitions.[57 58] As we could not identify a validated tool that was sufficiently brief, thyroid dysfunction is assessed with questions developed for the study based on Zulewski's list of symptoms of hypothyroidism[59] and fieldworker's visual assessment of goitre (bespoke protocol based on the WHO case definition of goitre[60]). Liver disease is assessed with a 2D ultrasound to detect liver fibrosis and fatty liver combined with fasting blood biomarkers (table 3). Chronic kidney disease is assessed by estimated Glomerular Filtration Rate based on serum creatinine, weight, age and sex using appropriate equations, and urine albumin to creatinine ratio. Gastro-oesophageal reflux disease is assessed with a symptom-based questionnaire developed for Indian populations.[61]

As recommended by a recent working group of experts in multimorbidity,[1] we are measuring additional states of poor health (poor oral health, chronic pain, frailty, chronic stress) as well as disability. Oral health is assessed with a symptom-based questionnaire of a nationwide survey in India.[62] We define chronic pain as persistent or recurring pain lasting more than 3 months (in line with the International Classification of Diseases 11th revision (ICD-11)), assessed with a questionnaire from the UK Biobank pain survey (as we could not identify a sufficiently brief questionnaire validated in India[63 64]). We conceptualise frailty with the widely used Fried frailty phenotype[65]: slowness (measured with the 6 meter timed walk), physical activity (validated APCAPS physical activity questionnaire[66]), weakness (grip strength), weight loss (question devised for the study (tuberculosis symptom questionnaire)) and exhaustion (question in the depression tool, Patient Health Questionnaire-9). The short (four-item) version of the cross-culturally validated Perceived Stress Scale is used to assess chronic psychological stress to minimise participant burden.[67] As recommended by prior studies in rural India, disability is measured with a validated functional limitation tool previously tested in India,[68 69] as well as screening for impairments. Vision impairment (near and distant) is measured with a standard visual

**Table 2** Measures collected in planned study (2022–23), source of questionnaire/instrument and protocol, and availability in prior (2010–2012) APCAPS study

| Measure | Source of questionnaire/Instrument and protocol | Availability in 2010–2012 |
|---|---|---|
| Questionnaires | | |
| Sociodemographics | APCAPS 2010–12 survey | ● |
| Food insecurity | Household Food Insecurity Access Scale (9-item)[80] | ○ |
| Family health history | APCAPS 2010–12 survey | ● |
| Tobacco use | APCAPS 2010–12 survey | ● |
| Alcohol use | Validated APCAPS food frequency questionnaire[81] | ● |
| Diet | Validated APCAPS food frequency questionnaire[81] | ● |
| Physical activity | Validated APCAPS physical activity questionnaire[66] | ● |
| Sleep | APCAPS 2010–2012 survey | ● |
| Medical history | APCAPS 2010–2012 survey, photos of medicines. | ● |
| CHD symptoms | WHO Rose Questionnaire[36] | ● |
| PAD symptoms | WHO Rose Questionnaire[36] | ○ |
| Stroke | Gourie-Devy et al, survey[37] | ⊖ |
| Cataracts | SAGE survey | ○ |
| Chronic obstructive pulmonary disease | Lung Function Questionnaire[44] | ⊖ |
| Asthma | European Community Respiratory Health Survey (short)[82] | ⊖ |
| Arthritis | SAGE survey | ○ |
| Depression | Patient Health Questionnaire-9[83] | ● |
| Generalised Anxiety Disorder | Generalised Anxiety Disorder-7[84] | ○ |
| Alcohol Use Disorder | Alcohol Use Disorders Identification Test[85] | ○ |
| Dementia | Brief Community Screening Instrument for Dementia[53] | ○ |
| Tuberculosis | Developed for study (Government of India's case definition of presumptive tuberculosis plus night sweats[54 55]) | ⊖ |
| COVID-19, post-COVID-19 condition | Developed for study (WHO case definitions)[57 58] | ○ |
| Hypothyroidism | Developed for study (Zulewski case definition[59]) | ⊖ |
| Gastro-oesophageal reflux disease | Indian Society of Gastroenterology survey[61] | ○ |
| Oral health | LASI survey | ○ |
| Chronic pain | UK Biobank survey[63] | ○ |
| Chronic stress | Perceived Stress Scale (Short Version)[86] | ○ |
| Disability | Washington Group Short Set on Functioning[87] | ⊖ |
| Health-related quality of life | EuroQol 5-Dimension Health Questionnaire, 5-level[88] | ● |
| Healthcare use | LASI survey | ○ |
| Medicine costs | LASI survey | ○ |
| Falls | Developed for study (Prevention of Falls Network Europe case definition[89]) | ○ |
| Cause of death | 2022 WHO Verbal Autopsy Instrument[90] | ○ |
| Physical/cognitive | | |
| Brachial blood pressure | Three readings taken in the right arm, sitting position (Omron HEM-7121 and Uscom BP+) | ● |
| Heart structure and function | Two-dimensional Echocardiogram (Philips CX-50 2D-ultrasound, S5-1 PureWave Cardiac Sector probe) | ○ |
| Arrhythmia | Resting ECG (12-lead 12-channel Cardiax ECG) | ○ |
| Carotid Intima–Media Thickness | Close to bifurcation (Philips CX-502D-ultrasound, linear probe, Philips QLAB software) | ⊖ |

Continued

**Table 2** Continued

| Measure | Source of questionnaire/Instrument and protocol | Availability in 2010–2012 |
|---|---|---|
| Arterial stiffness | Three readings taken in the right arm, sitting position (Uscom BP+suprasystolic central blood pressure monitor) | ⊖ |
| Standing height | End of expiration to nearest 1 mm (Seca Leicester stadiometer) | ● |
| Weight | Single frequency Bioelectrical Impedance (BIA) Scale (TANITA BC418 M57NA) or, if unable to undergo BIA, digital scales to the nearest 0.1 kg (Seca 899) | ● |
| Body composition | Single frequency BIA scale (TANITA BC418 M57NA), two measures taken wearing light clothes and bare feet. | ● |
| Waist circumference | Narrowest point of the waist from the front (ADE non-stretch metallic tape) | ● |
| Hip circumference | Maximum extension of buttock from the side (ADE non-stretch metallic tape) | ● |
| Diabetic ulceration | Infrared images of both feet following 15 min rest (CAT-S61 smartphone with FLIR thermal imaging camera) | ○ |
| Diabetes neuropathy | Sensory assessment of the foot and symptom-based questionnaire (mTCNS)[39] | ○ |
| Toe brachial index | Ratio of brachial and toe pressures (three measures), photoplethysmography method (Vicorder, Skidmore Medical), taken in a supine position after 5 min rest. | ○ |
| Retinopathy | Retinal images centred on the optic nerve and macula taken in a darkened room without dilation (Bosch digital fundus camera (non-mydriatic)) | ○ |
| Gait speed and characteristics | 6 meter timed walk at normal speed with dynamic start and no deceleration with a waist-worn accelerometer (Actigraph) | ○ |
| Grip strength | Standing position, elbows at 20° angle, two measures in both hands (Lafayette Hand-held Dynamometer 78010) | ● |
| Goitre | Visual assessment (WHO goitre case definition)[60] | ○ |
| Liver fibrosis | Ultrasound of right lobe with periportal vasculature visible (Philips CX-50 2D-ultrasound, C5-1 PureWave Convex probe) | ○ |
| Vision impairment | Tumbling E LogMAR chart (4 m and 40 cm distance) (Precision Vision) | ○ |
| Hearing impairment | Phone-based pure-tone audiometry (Hearing Test (pro) application (e-audiological.pl),[91] Sennheiser HD 458 BT headphones) | ○ |
| Retrieval fluency | Maximum animals named in 1 min (animal naming test) | ○ |
| **Environment** | | |
| Village-level urbanisation | Night-time light intensity remote sensing data[92] | ● |
| Air pollution | PM2.5 and black carbon prediction models[93] | ● |
| Built environment | Shops selling food, tobacco and alcohol, health services, educational services, advertisements, public transport and walkability, physical activity (surveyed and geolocated) | ⊖* |

●: collected with same instrument, ⊖: collected with different instrument, ○: not collected in 2010–2012.
*Surveyed and geolocated in 2015-16, not updated in 2022–2023.
APCAPS, Andhra Pradesh Children and Parents Study; CHD, Coronary Heart Disease; FLIR camera, Forward-looking infrared camera; LASI, Longitudinal Aging Study in India; mTCNS, modified Toronto Clinical Neuropathy Score; PAD, Peripheral Arterial Disease; PM2.5, particulate matter 2.5; SAGE, Study on Global Ageing and Adult Health.

acuity test protocol using the Tumbling E LogMAR chart, which is appropriate for people with low literacy. Hearing impairment is assessed with a smartphone-based hearing threshold test using the validated Hearing Test (pro) application (e-audiological.pl).[70] HearingTest has demonstrated good validity at higher frequencies[71 72]; presbycusis (characterised by hearing loss at high frequencies) is likely the most common cause of hearing loss in the

**Table 3** Biological samples collected in planned study (2022–2023), protocol/assay and availability in prior (2010–2012) APCAPS study

| Stored biological samples and assays | | |
| --- | --- | --- |
| **Measure** | **Protocol/assay** | **Availability in 2010–2012** |
| Blood | Venous blood (overnight fast), processed on the same day (plasma and serum aliquots) and stored at −80°C | ● |
| Urine | First urination of the day, mid-stream, processed on the same day and stored at −80°C | ○ |
| Haemoglobin | Auto-analyser (ABX Micros 60-HORIBA) | ⊖ |
| Fasting blood glucose | GOD-POD method (Randox laboratories, India) | ● |
| HbA1c | To be confirmed | ○ |
| Serum lipids (HDL, TC, Trig) | To be confirmed | ● |
| Liver function tests (ALT, AST, ALP, bilirubin) | To be confirmed | ● |
| Serum creatinine | To be confirmed | ● |
| Urine albumin to creatinine ratio | To be confirmed | ○ |

●: collected with same instrument, ⊖: collected with different instrument, ○: not collected in 2010–2012.
ALP, alkaline phosphatase; ALT, alanine transaminase; APCAPS, Andhra Pradesh Children and Parents' Stud; AST, aspartate aminotransferase; HbA1c, glycated haemoglobin; HDL, high-density lipoprotein; TC, total cholesterol; Trig, triglycerides.

study population.[73] Information on deaths occurring in the cohort will be collected from Accredited Social Health Activist (ASHA) workers, who routinely monitor mortality in the villages, on a 6-monthly prospective basis (project funding permitting). Close contacts of the deceased will be interviewed using the validated WHO verbal autopsy instrument (2022 version) to ascertain cause of death.[74]

## Sample collection and processing
Participants are asked to fast overnight (at least 8 hours) and rest quietly for 10 min before venous blood samples (up to 20 mL) are collected from their preferred arm by a trained phlebotomist (table 3). Blood is collected into appropriate vacutainers (two plain, one EDTA, one Na-F EDTA, Becton Dickinson, New Jersey, USA) then centrifuged and aliquoted in situ within 1–4 hours. Samples are kept on ice until being transferred to −80°C freezers at the central laboratory, within 6 hours of collection. Participants are given a urine collection container the day before their scheduled clinic and instructed to collect a mid-stream urine sample from their first urination the next morning. Samples are kept on ice during the clinic and transferred to the central laboratory to be centrifuged (to remove cells/debris), aliquoted and stored at −80°C that day.

## Sample analysis
On the same day as sample collection, the following assays are being conducted at the central laboratory (ICMR-NIN, Hyderabad, India): Complete blood count using autoanalyser (Horiba ABX Micros 60 OT, Japan) and plasma glucose using GOD-POD method (Randox laboratories India) (table 3). Appropriate 2-level or 3-level quality controls are in place as provided by the

manufacturers. Other assays will be conducted in batches using frozen serum or plasma or urine samples.

## Quality of quantitative data
Fieldworkers were trained in consent procedures, ODK, survey interviewing and the anthropometry examination (following the same protocols as prior APCAPS surveys) by the experienced APCAPS supervisory team (10–15 years working with the cohort). For measures not collected in prior studies (table 2), fieldworkers were either trained by coinvestigators (in their respective fields) or external experts. Developers of the mTCNS instrument and a neurophysician trained fieldworkers to conduct the neuropathy assessment.[39] A clinical ophthalmologist trained fieldworkers to collect retinal images, provided quality control feedback on initial images collected and refresher training. Experienced technicians trained in 2D-echo are undertaking the echocardiogram, liver scan and CIMT. A team of experienced phlebotomists and biotechnicians are collecting, aliquoting and storing biological samples. Prior to data collection, the full protocol (questionnaires and physical examinations) was piloted in the community.

To maximise reliability, rigorous protocols have been developed, each measure is collected by the same 2–3 fieldworkers to minimise inter-rater variability, and instruments are calibrated at daily, weekly, monthly and/or yearly intervals based on relevant guidelines. The supervisory team observe, evaluate and feedback on protocol implementation in the field using a quality control proforma. To minimise data errors, data collected through the ODK platform restricts implausible values, and other data are extracted directly from instruments when feasible. Data quality reports that describe missing

data, variable distributions and questionnaire completion times are assessed by the research team fortnightly. On an ongoing basis, quality of echocardiograms, ECG traces and CIMT images are examined by an experienced cardiologist and the retinal images with an automated tool.[75] The protocol is repeated in a random 5% of participants 2–4 weeks following initial data collection, intraclass correlation coefficients (ICCs) are estimated for key continuous variables (eg, anthropometrics) as the study is ongoing, and refresher training conducted if ICCs indicate decreasing reliability.

To ensure correct data linkage, participants' personal details are used to link them to their unique APCAPS identifiers (ID) on attending the clinic. Each participant has a folder with a QR code linked to their APCAPS ID to track protocol completion, minimise missing data, link each dataset to the correct participant and link incoming data to existing APCAPS data. Sample vials are labelled with QR codes linked to APCAPS IDs. During data cleaning, age and sex will be cross-checked across datasets, and discrepancies will be compared against raw data to ensure correct linkage.

## Data analysis

While the study team have prespecified analyses, external researchers can use APCAPS data to assess their own research questions[25] (see data sharing). Assessment of the burden of multimorbidity will include descriptive statistics of the prevalence and (where possible) 10-year cumulative incidence of multimorbidity. Multimorbidity outcomes will be defined as both count and clusters of conditions (our primary outcome), conjointly identified with two commonly used clustering methods (latent class analysis and multiple correspondence analysis.)[76] Risk factors (eg, environmental, sociodemographic, behavioural and biological) and consequences (eg, health-related quality of life) of multimorbidity will be assessed by multiple logistic or ordinal logistic regression modelling as appropriate (figure 1), adjusted for appropriate covariates and using multiple imputation to complete missing data values where necessary.

## Limitations

Comparable baseline data for incidence analyses will be lacking for some conditions. However, for several age-related conditions (eg, dementia, sarcopenia) this is unlikely to bias the incidence estimates notably due to their low prevalence in under 50s (mean cohort age 49 years in 2010–12). Other conditions that may have manifested at baseline (eg, generalised anxiety disorder) can be excluded from incidence analyses. Self-reported clinical diagnoses will not be validated against medical notes due to inconsistent record keeping by participants. To reduce participant burden, screening instruments were selected to balance validity in the study population and time taken to complete. As a result, some conditions were not assessed with gold-standard measures (eg, tuberculosis).

## Qualitative data collection

Following quantitative data collection, we will use qualitative methods to explore the lived experience of individuals with multimorbidity and their informal carers, to understand how healthcare is organised for people with multimorbidity, and to identify the barriers and enablers to high-quality care of people with multimorbidity. Participants will include individuals with multiple diagnosed conditions, their informal carers (defined as someone who helps them with their health regularly), and local healthcare providers involved in the diagnosis and management of chronic conditions. This is guided by the burden of treatment theory, which models how patients, their social networks and the healthcare system interact to address the burden of treatment, a key issue in multimorbidity.[19 77]

## Qualitative sample

We will use a purposive sampling strategy aiming for maximum variation in clusters of diagnosed conditions. The quantitative data will be analysed using clustering approaches to identify the most common concordant, discordant and complex multimorbidity clusters (approximately N=15).[78] Around half of the interviewees will be asked to refer an informal carer to participate. We will also aim for maximum variation in healthcare provider type, including doctors in local primary health centres and district hospitals, registered medical practitioners and pharmacists (approximately N=15). Transcripts will be reviewed as data collection is ongoing, further interviews may be undertaken to explore issues being raised by participants.

## Qualitative methods and analysis

We will undertake in-depth, one-to-one interviews with participants in the local language (Telugu), which will be audiorecorded. The burden of treatment theory, plus existing evidence on diagnosis, management and impact of chronic conditions in India, will guide development of the topic guides.[77] We will use a reflective thematic analysis approach which is appropriate for capturing diverse experiences and developing practice-relevant recommendations.[79] We will undertake inductive-deductive coding, guided but not restricted to a coding framework with high-level categories based on the burden of treatment theory.[77]

## Qualitative data quality

Prior to data collection, topic guides will be piloted in the community to evaluate relevance, comprehension and length. Interviews will be conducted by experienced qualitative interviewers. To be sensitive to local gender norms and maximise participant comfort, interviewers of the same gender as participants will undertake the interviews in private locations (as feasible) of the participants' choosing. A subset of the early transcripts will be double-coded to clarify code definitions and initiate discussion and interpretations.

## Patient and public involvement

The study design was based on insights from in-depth interviews and focus groups with community members and local health providers, who perceived a high burden of a range of cardiometabolic, mental health, and skeletal conditions, and sensory and mobility impairments in the population.[27] The study design was discussed with and approved by village leaders.

## Ethics

The study protocol and tools have been approved by the ethics committees of ICMR-NIN (CR/1/V/2023) and Indian Institute of Public Health Hyderabad (IIPHH/TRCIEC/189/2018), India, and the London School of Hygiene and Tropical Medicine (21771/RR/19113), UK. Trained fieldworkers will verbally explain the study information sheet to potential participants; if willing to take part, participants will be asked to sign the consent form or thumbprint with a witness present if they are not literate. To maximise the safety of participants and study staff, participants with current household cases or symptoms of COVID-19 will be temporarily ineligible to participate and will be requested to revisit the clinic after 21 days. Strict hygiene protocols will be followed and data collection will be paused during periods of high COVID-19 infection. Physical assessments for women will be undertaken by female fieldworkers or with a woman present. Using objective measures to screen for chronic conditions may result in high cognitive and time burden for participants (full clinic protocol approximately 4 hours). To reduce this, participants will be provided meals and refreshments, questionnaires will be administered earlier in the day, and participants will be reimbursed for their time. All participants will be provided with a basic report detailing their BMI, blood pressure, fasting blood glucose and haemoglobin results. Abnormal physical assessment or biochemical results will be explained to participants by the APCAPS medical officer, who will refer participants to local government or charitable healthcare services for confirmatory diagnoses and treatment.

## Dissemination

We will publish our findings in open-access and peer-reviewed journals and disseminate them through scientific conferences and social media accounts of the APCAPS cohort (twitter: @apcaps_study) and team members. We will share summaries of our findings with village leaders, the local community and the state health department.

## Data sharing

We strongly encourage external researchers to use APCAPS data for multimorbidity or other health research. Data will be held under embargo for the study team to conduct the above analyses for 1 year following the database lockdown. Thoroughly deidentified quantitative and qualitative data will be made available to external researchers following approval of a collaboration request form by the APCAPS executive committee. Meta-data and data collection instruments will be published on the APCAPS website alongside details of existing APCAPS data (www.lshtm.ac.uk/research/centres-projects-groups/apcaps) and disseminated via social media and networks of the research team. Please contact apcaps.crf@gmail.com with data queries.

**Author affiliations**
[1]Department of Non-communicable Disease Epidemiology, London School of Hygiene and Tropical Medicine Faculty of Epidemiology and Population Health, London, UK
[2]National Institute of Nutrition, Hyderabad, Telangana, India
[3]Public Health Foundation of India, New Delhi, India
[4]Nagasaki University, Nagasaki, Japan
[5]Centre for Global Chronic Conditions, Faculty of Epidemiology and Population Health, Department of Non-communicable Disease Epidemiology, London School of Hygiene & Tropical Medicine, London, UK
[6]Coventry University, Coventry, UK
[7]Indian Institute of Public Health Hyderabad, Hyderabad, India
[8]SACDIR, Public Health Foundation of India, New Delhi, India
[9]International Center for Evidence in Disability, London School of Hygiene & Tropical Medicine, London, UK
[10]Enact Health Private Limited, Hyderabad, India
[11]Indian Council of Medical Research, New Delhi, India

**Acknowledgements** We would like to acknowledge and thank the various researchers and practitioners in each partner institution and Ranga Reddy healthcare system that advised on elements of the study protocol, village leaders and local stakeholders for supporting and facilitating data collection, ICMR-NIN for permitting use of their facilities for data collection, the hard-working field team and most imperatively, the participants who kindly gave us their time.

**Contributors** JL, SKB, PACM, HM, OK, SS, BK and SK were involved in conception of the study. JL, SKB, PACM, HM, SB, SA, NB, WS, ASVS, OK, GI, KS, RKG, SS, TD, BK and SK were involved in design and development of the study protocol. JL and SKB contributed equally to developing the manuscript. All authors substantially contributed and commented on the manuscript and approved the final version.

**Funding** This work was primarily supported by the UK Medical Research Council (MRC) grant number MC_PC_MR/T038292/1. This research was also part funded by MRC grant number MR/V001221/1, the Nagasaki University 'Doctoral Program for World-leading Innovative and Smart Education' for Global Health, KENKYU SHIDO KEIHI and in-kind support from ICMR-NIN.

**Competing interests** None declared.

**Patient and public involvement** Patients and/or the public were involved in the design, or conduct, or reporting, or dissemination plans of this research. Refer to the Methods section for further details.

**Patient consent for publication** Not applicable.

**Provenance and peer review** Not commissioned; externally peer reviewed.

**ORCID iDs**
Judith Lieber http://orcid.org/0000-0002-8514-0381
Poppy Alice Carson Mallinson http://orcid.org/0000-0002-7591-8065
Wenbo Song http://orcid.org/0000-0002-1588-8597
Sureshkumar Kamalakannan http://orcid.org/0000-0003-4407-7838
Bharati Kulkarni http://orcid.org/0000-0003-0636-318X

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
