## [Reviewer comments · BMJ Open]

ARTICLE DETAILS

TITLE (PROVISIONAL)	Burden, determinants, consequences, and care of multimorbidity in rural and urbanising Telangana, India: Protocol for a mixed-methods study within the APCAPS Cohort
AUTHORS	Lieber, Judith; Banjara, Santosh Kumar; Mallinson, Poppy; Mahajan, Hemant; Bhogadi, Santhi; Addanki, Srivalli; Birk, Nick; Song, Wenbo; Shah, Anoop; Kurmi, Om; Iyer, Gowri; Sureshkumar, K; Kishore Galla, Raghu; Sadanand, Shilpa; Dasi, Teena; Kulkarni, Bharati; Kinra, Sanjay

VERSION 1 – REVIEW

REVIEWER	Lall, Dorothy Institute of Public Health, Health services
REVIEW RETURNED	01-Jun-2023

GENERAL COMMENTS	Thank you for the manuscript of a protocol that seeks to study multi morbidity through a cohort design. The manuscript is well written and clearly presented. Just a few concerns to consider 1. The data collection instruments and definitions of disease proposed for this round are not consistent with previously collected data - how will meaningful comparisons be made and how can incidence be determined. Please add a paragraph on limitations2. Analysis especially quantitative can be elaborated - more clarity on what are the primary outcomes of interest and what factors will be considered for the regression model proposed3. Qualitative analysis also needs some more detailing, how will codes be decided, what is the approach being used- if phenomenological then appropriate analysis should be mentioned
---

VERSION 1 – AUTHOR RESPONSE

1. The data collection instruments and definitions of disease proposed for this round are not consistent with previously collected data - how will meaningful comparisons be made and how can incidence be determined.

Please add a paragraph on limitations

Thank you for your suggestion; we strongly agree with the importance of ensuring data is consistent across surveys for incidence analyses. Comparable baseline data is available for several key conditions, for example hypertension, diabetes, depression, anaemia, coronary heart disease,

underweight, and overweight/obesity. For some conditions, we collected more biomarkers than in previous surveys (e.g., liver disease) or used a more extensive version of the baseline questionnaire (e.g., asthma). We can easily standardise this data to ensure comparability, for example restricting the asthma case definition to questions common to both surveys. For chronic-obstructive pulmonary disease, we used a symptom-based questionnaire tool instead of spirometry as we wished to minimise the risk of COVID-19 transmission. Baseline data is lacking for several age-related conditions (e.g., dementia, cataracts, sarcopenia, left-ventricular dysfunction, stroke, atrial fibrillation). We did not screen for these age-related conditions in previous surveys as we expected low prevalence due to age of the cohort (mean age 49 years in 2010-12). For incidence analyses, researchers could assume these conditions had not yet manifested at baseline and potentially restrict the sample (e.g., removing cohort members aged 55-plus in 2010-12) to strengthen this assumption. Baseline data is unavailable for a handful of conditions which may have manifested at younger ages (generalised anxiety disorder, alcohol use disorder, gastro-oesophageal reflux disease). We selected to measure these in the current study to expand the research focus to multimorbidity; previous APCAPS surveys largely focused on cardiometabolic disease. These conditions would likely need to be excluded from multimorbidity incidence analyses. We have now added a paragraph to highlight this and other limitations, and to suggest an approach for maximising comparability.

“Comparable baseline data for incidence analyses will be lacking for some conditions. However, for several age-related conditions (e.g., dementia, sarcopenia) this is unlikely to bias the incidence estimates notably due to their low prevalence in under-50s (mean cohort age 49 years in 2010-12). Other conditions that may have manifested at baseline (e.g., generalised anxiety disorder) can be excluded from incidence analyses. Self-reported clinical diagnoses will not be validated against medical notes due to inconsistent record keeping by participants. To reduce participant burden, screening instruments were selected to balance validity in the study population and time taken to complete. As a result, some conditions were not assessed with gold-standard measures (e.g., tuberculosis).”

2. Analysis especially quantitative can be elaborated - more clarity on what are the primary outcomes of interest and what factors will be considered for the regression model proposed

We initially did not extensively detail the study team's planned analyses (e.g., defining primary exposures and outcomes) as the data resource will be made available for the wider research community and we did not wish to limit the scope of analyses by external researchers. However, we appreciate that it could be useful to provide additional information on potential analyses. We have edited the quantitative analysis section to specify the study team's primary outcomes and planned analytical methods and to highlight the range of risk-factors that could be considered .

“While the study team have prespecified analyses, external researchers can use data from this and other APCAPS surveys to assess their own health-related research questions (25) (see data sharing). Assessment of the burden of multimorbidity will include descriptive statistics of the prevalence and (where possible) 10-year cumulative incidence of multimorbidity. Multimorbidity outcomes will be defined as both count and clusters of conditions (our primary outcome), conjointly identified with two commonly used clustering methods (latent class analysis (LCA) and multiple correspondence analysis (MCA)). Risk factors (e.g., environmental, sociodemographic, behavioural, and biological) and consequences (e.g., health-related quality of life) of multimorbidity will be assessed by multiple logistic or ordinal logistic regression modelling as appropriate (figure 1), adjusted for appropriate covariates and using multiple imputation to complete missing data values where necessary.”

3. Qualitative analysis also needs some more detailing, how will codes be decided, what is the approach being used- if phenomenological then appropriate analysis should be mentioned

Similar to the quantitative analysis section, we initially did not detail our qualitative analysis plan to avoid restricting the scope of qualitative analyses conducted by external researchers. For example, while we will take a reflective thematic analysis approach, other researchers could restrict the sample to participants with one multimorbidity cluster to take an interpretive phenomenological analysis approach. We have now further detailed the study team's qualitative analysis plans.

"We will undertake in-depth, one-to-one interviews with participants in the local language (Telugu), which will be audio-recorded. The Burden of Treatment Theory, plus existing evidence on diagnosis, management, and impact of chronic conditions in India, will guide development of the topic guides (90). We will use a reflective thematic analysis approach which is appropriate for capturing diverse experiences and developing practice-relevant recommendations (92). We will undertake inductive-deductive coding, guided but not restricted to a coding framework with high-level categories based on the Burden of Treatment Theory (90)."